# Impact of Lactation Stage on the Metabolite Composition of Bovine Milk

**DOI:** 10.3390/molecules28186608

**Published:** 2023-09-14

**Authors:** Claire Connolly, Xiaofei Yin, Lorraine Brennan

**Affiliations:** 1UCD School of Agriculture and Food Science, UCD Institute of Food and Health, University College Dublin, 4 Dublin, Irelandxiaofei.yin@ucd.ie (X.Y.); 2UCD Conway Institute of Biomolecular and Biomedical Research, University College Dublin, 4 Dublin, Ireland

**Keywords:** lactation stage, bovine milk, metabolomics

## Abstract

Bovine milk is a nutrient-dense food and a major component of the human diet. Therefore, understanding the factors that impact its composition is of great importance. Applications of metabolomics provide in-depth analysis of the metabolite composition of milk. The objective of this research was to examine the impact of lactation stage on bovine milk metabolite levels. Metabolomic analysis of bovine milk powder samples across lactation (N = 18) was performed using nuclear magnetic resonance (^1^H-NMR) spectroscopy and liquid chromatography–tandem mass spectrometry (LC-MS/MS). Forty-one metabolites were identified and quantified in the ^1^H-NMR spectra. Statistical analysis revealed that 17 metabolites were significantly different across lactation stages (FDR < 0.05), of which the majority had higher levels in early lactation. In total, 491 metabolites were measured using LC-MS/MS, of which 269 had significantly different levels across lactation (FDR < 0.05). Compound classes significantly affected by lactation stage included phosphatidylcholines (59%) and triglycerides (64%), of which 100% of phosphatidylcholines and 61% of triglycerides increased from early lactation onwards. Our study demonstrates significant differences in metabolites across the stages of lactation, with early-lactation milk having a distinct metabolomic profile. More research is warranted to further explore these compositional differences to inform animal feeding practice.

## 1. Introduction

The consumption of bovine milk and associated dairy products is estimated at 6 billion people globally [1]. While milk consumption in Western societies has declined in recent years, consumption in developing countries has doubled, and world milk production is expected to grow 1.8% per annum by 2028 [1,2]. In countries such as the United States, the dairy industry has greatly evolved in recent decades, with smaller family farms being replaced by larger factory farms, impacting both the composition and quantity of the milk produced [3,4]. Milk is a rich source of key nutrients for the human population, and consequently, understanding the factors impacting its composition and physiochemical properties is crucial for the dairy industry. 

Milk contains essential macronutrients such as water (85–87%), fats (3.8–5.5%), proteins (2.9–3.5%), carbohydrates (5%) and a range of vitamins, minerals, oligosaccharides, immunoglobulins and various lipids [5]. Nutrient composition varies due to endogenous factors such as stage of parity, breed, stage of lactation and exogenous factors including dietary regime and seasonality [6,7,8,9]. Variations in nutritional composition impact the processing properties of milk and the quality of dairy products produced. A well-established factor affecting the composition of milk is cow breed, with fat and protein content of milk differing between cow breeds. For example, Brown Swiss breeds produce milk containing higher fat and protein yields compared to Holsteins [10]. Additionally, alterations in gross milk composition occur due to dietary feeding regimes, with outdoor pasture-based cattle producing milk containing higher concentrations of total solids, protein and casein and total-mixed ration (TMR) feed regimes containing higher lactose concentrations [10,11]. There is a growing body of evidence highlighting the pivotal role stage of lactation has in milk composition. As a result of their temperature climates, countries such as Ireland and New Zealand employ pasture-based seasonal calving systems, which leads to synchronized variations in milk composition depending on which lactation stage the milk is obtained in [8,12]. These feeding regimes account for 10% of global milk supplies, with TMR feed regimes being more common due to increased demand for global milk supplies and their beneficial effect on milk yield [7]. Physiological changes associated with seasonal systems cause alterations in milk composition, milk quality and quantity [8,11,12,13,14]. Macronutrients including fat, protein, total solids, casein and whey increase across lactation irrespective of whether a pasture or concentrate-based regime is employed, whereas lactose concentrations decrease and micronutrients such as calcium fluctuate with lactation, similar to casein content, due to storage in casein micelles [8,11,12,14,15]. Minerals such as calcium play a key role in protein aggregation such as rennet coagulation in cheese production [11]. Therefore, these results highlight the key impact of lactation stage on both the macro- and micronutrient composition of bovine milk and demonstrate its pivotal role in milk processing and the production of downstream dairy products.

Metabolomics measures small molecules in biological samples, enabling detailed characterization of samples such as milk [15,16,17]. Analysis of milk metabolites using different metabolomic tools has identified over two thousand metabolites, enhancing knowledge regarding factors affecting bovine milk composition and contributing to the authentication of milk from pasture-based regimes [18]. For example, late-lactation milk metabolic profiles from Friesian and autochthonous cows were distinctly different between breeds. Friesian cows had lower levels of the metabolites acetylcarnitine, carnitine and fumarate, and autochthonous breeds demonstrated higher levels of ribosyl and cytidine metabolites [19]. Regarding parity, metabolomics discriminates between colostrum and transition milks of first-, second- and third-parity cows, which is attributed to the higher levels of conjugated linoleic acid in cows on their first lactation, and higher levels of C16:0 FA in multiparous cows [10,20,21]. Moreover, metabolomic analysis demonstrated how different feeding regimes impact the amino acid profile of skim milk powder, with levels of metabolites including glutamine, valine and phosphocreatine being higher in TMR feeding regimes, highlighting the effect of external factors on milk [22]. These findings provide insight into the impact of breed, parity and dietary regime on the metabolomic profile of milk, its associated products and the animal themselves. For the most part, studies investigating lactation focus on early lactation and negative energy balance (NEB), a metabolic stress which occurs when energy demands outweigh energy intake [23]. The metabolites acetone and β-hydroxybutyrate were suggested as candidate biomarkers for cow energy status metabolism due to the significantly higher levels found in early lactation, which decrease thereafter [16,24]. Furthermore, metabolomic analysis identified twenty metabolites negatively associated and fifteen positively associated with cow energy balance, highlighting differences occurring in the milk metabolome in early lactation [25]. With respect to mid- and late-lactation milk, multivariate analysis of the milk metabolome identified key metabolites responsible for the separation between these lactation stages, including 1,2-propandiol, 3-hydroxybutyrate, butyrate, N-acetyl-X4, galactose-1-P, glucose-1-P, N-acetyl-glucosamine and cytidine-X-P [19]. Although several studies have examined alterations in the metabolomic profile of milk and dairy products concerning parity, diet and lactation, the impact of lactation in a pasture-based dietary system is not fully understood. In recent decades, consumer preference for pasture-fed milk and dairy products has grown due to the perceived enhanced health benefits and positive association with animal welfare. Therefore, gaining insight into alterations in the metabolomic profile across lactation is imperative to enhance nutritional support, milk processing and dairy product production. Therefore, the objective of this study was to characterize the impact of lactation on the metabolite composition of grass-fed bovine milk. 

## 2. Results

### 2.1. Gross Compositional Analysis

The macronutrient concentration in the milk powder samples was stable across the stages of lactation (Table 1). Changes in the mineral composition were evident with the highest calcium levels in early-lactation milk samples, with similar trends evident for phosphorus, sodium and zinc.

### 2.2. Metabolomic Data Analysis

#### 2.2.1. ^1^H-NMR Data Analysis

Metabolomic profiling of the milk samples was carried out using a comprehensive, multiplatform approach including ^1^H-NMR and LC-MS/MS technologies, covering a total of 532 metabolites. In total, 41 metabolites were profiled in milk powder samples using ^1^H-NMR, which are presented in Table 2. Examination of these metabolites revealed that the early-lactation milk was distinctly different to mid- or late-lactation samples. A total of 17 metabolites were significantly different across the lactation stages (FDR < 0.05). Metabolites including creatinine phosphate, o-phosphocholine, creatine, citrate, dimethylamine, glutamate, 3-hydroxybutyrate, glycerophosphochline and d-maltose decreased as lactation progressed. Metabolites such as choline, glucose-1-phophate, hipurate, dimethylsulfone, galactose, ethanolamine, n-acetylgalactosamine and orotic acid increased with lactation stage.

#### 2.2.2. LC-MS/MS Data Analysis

Using LC-MS/MS, a total of 491 metabolites from various compound classes, including acylcarnitines (N = 10), amine oxides (N = 1), amino acids (8), amino acid-related compounds (N = 17), bile acids (N = 6), biogenic amines (N = 3), carboxylic acids (n = 5), ceramides (N = 25), cholesterol esters (N = 20), diglycerides (N = 32), dihexosylceramides (N = 8), fatty acids (N = 8), hexosylceramides (N = 16), lysophospholipids (N = 8), phospholipids (N = 59), sphingomyelins (N = 15), triglycerides and choline, were analysed. Metabolites with the highest levels in the milk samples included triglycerides such as TG (18:1_34:1), TG (18:1_32:1), TG (18:1_36:2), TG (18:1_32:0) and TG (16:0_34:1). One-way ANOVA analysis revealed that a total of 269 metabolites were significantly different across the three lactation stages (FDR ≤ 0.05). The overall impact of lactation on the metabolite profile is evident in the heatmap, with a distinct pattern of higher metabolite levels observed in early lactation (Figure 1). Stage of lactation affected all compound classes, having a particular impact on certain classes, including triglycerides and phospholipids (Figure 2). In total, 239 triglycerides were measured, 153 of which significantly changed across lactation, with 94 increasing and 59 decreasing as lactation progressed. Analysis of these trends determined that triglycerides with increased levels commonly had an acyl chain length of 14 or 16 carbons in one of the triglyceride sidechains (Figure 2). Similarly, triglycerides with a fatty acid of 18 carbons decreased from early to late lactation (Figure 2). Moreover, triglyceride composition changed significantly across lactation stage with respect to total carbon number (CN) and number of double bonds (DBs). Triglycerides of a higher carbon number (CN56, CN55, CN54, CN53, CN52, CN51 and CN50) tended to decrease in concentration across lactation, whereas triglycerides with a lower molecular weight (CN49, CN48, CN46 and CN44) increased (Figure 3). With respect to other lipid classes, 32 phosphatidylcholines and 6 lysophospholipids significantly increased across lactation (FDR < 0.05). Significant phosphatidylcholines were composed of medium and long acyl chain lengths, both saturated and unsaturated. Similarly, the majority of significant diglycerides and cholesterol esters increased across lactation. The highest choline levels were identified in late-lactation milk, with significant differences in levels observed between early and mid and early and late lactation (FDR ≤ 0.05). Moreover, amino acids such as aspartate, cysteine, glycine and serine significantly decreased as lactation progressed (FDR ≤ 0.05).

## 3. Discussion

The purpose of this study was to investigate the alteration in the metabolomic profile of grass-based bovine milk powder across a full lactation season. Using a multiplatform metabolomic approach, the results indicate that the metabolomic profile of early-lactation cow’s milk is distinctly different from that of mid- or late-lactation milk. Understanding the alterations occurring in the composition of bovine milk across lactation is imperative for the agri-food industry due to the impact on downstream dairy product production. Milk composition is influenced by a range of factors including parity, genetics and cow breed and exogenous factors such as dietary regime and lactation stage. Seasonal variations in milk composition are well established, with studies highlighting that seasonal trends occur irrespective of dietary regime [8,14]. 

The results identify a clear modulation in the lipid profile of bovine milk across lactation. Well-established alterations occur in gross fat concentrations across lactation in pasture-based regimes, with gross fat concentrations increasing as lactation progresses [14]. The lipid fraction of milk is complex, accounting for 3–5% of overall milk gross composition, including several lipid classes, such as phospholipids, cholesterol and triglycerides. Prompt alterations in the milk lipidome as a result of lactation stage are well documented [14]. In this study, lactation stage had a clear impact on specific lipid classes, such as phospholipids and triglycerides. Bovine milk is an important source of polar lipids for the human population, with phospholipids accounting for 0.5–1% of milk lipids [26]. Phosphatidylcholines are the most abundant phospholipid class within bovine milk, playing a pivotal structural role in milk fat. Therefore, understanding their modulation across lactation is of great importance to the dairy industry, particularly for the formation of infant formula [27]. The results of this study indicate that phosphatidylcholine levels significantly increased from early to late lactation. In accordance with the present results, previous studies demonstrated that bovine milk collected in June and July contained a higher percentage of phospholipids compared to milk collected in March–May and September–November [28,29]. Furthermore, previous studies reported that the levels of phosphatidylcholines also increased across lactation [29,30]. Changes occurring in phospholipid levels across lactation are relative to changes in FA concentrations within milk, with lactation stage impacting FA synthesis rates as a result of an NEB [30]. Moreover, in this study, triglyceride composition was significantly impacted by lactation, with levels of 64% of triglycerides measured significantly changing across lactation. Variations in the triglyceride composition of milk led to alterations in physiochemical properties such as melting point, milk quality, milk processability and yield of associated dairy products such as cheese and butter. Interestingly, not all triglycerides changed in the same way across lactation: the high-molecular-weight triglycerides tended to decrease across lactation, while the medium-molecular-weight triglycerides increased across lactation. This result is in accordance with a study by Pacheco-Pappenheim et al. 2022, where similar trends were observed [31]. Unlike our study, the cows in the Pacheco-Pappenheim et al. study consumed a diet consisting of grass sileage, maize sileage and concentrates, demonstrating how lactational differences occur in bovine milk composition irrespective of dietary regime [31]. In summary, our targeted metabolomic approach provides novel insights into the impact of lactation stage on specific triglycerides. Combined with the phospholipid data, this provides a comprehensive overview of changes occurring during lactation. 

In addition to lipids, several other metabolite classes were significantly impacted by lactation stage. The current study found a significant increase in choline and a decrease in citrate and o-phosphocholine levels (FDR < 0.05). The results of this study indicate early-lactation milk had higher levels of citrate, an intermediate metabolite which is involved in cellular energy metabolism [32]. These results are consistent with Garnsworthy et al. (2009), who found citrate levels in early lactation were significantly different to those found in mid or late lactation, with no significance reported between mid- and late-lactation citrate levels. The researchers hypothesized that elevated citrate levels in early lactation are associated with an NEB, with metabolomic analysis correlating milk fat yield and citrate levels, suggesting higher citrate levels are related to de novo fatty acid synthesis in cows with an NEB, independent of feeding regime [32,33]. Moreover, these results suggest an improved energy balance between mid and late lactation due to the correction of the NEB [33]. Citrate is an important constituent of cow’s milk due to its role in milk quality and processing properties in downstream cheese production; therefore, understanding its variation across lactation is crucial for dairy processing [34]. Another interesting finding was the significant decrease in 3-hydroxybutyrate between early- and mid- and early- and late-lactation milk samples. As mentioned previously, 3-hydroxybutyrate is suggested as a candidate biomarker for cow metabolic status during an NEB, with elevated levels in early lactation compared to late lactation [16]. Unlike our study, this research only investigated metabolite differences between early- and late-lactation milk, where cows were not fed a purely pasture-based dietary regime [16]. Like the previous study, lactose levels did not significantly differ (*p* < 0.05) across lactation and remained relatively constant [14]. Milk and dairy products are an important source of choline in the human diet. Choline is an essential nutrient in human nutrition, being required for the formation of membrane phospholipids. In this study, choline levels increased across lactation, whereas o-phosphocholine levels decreased. This outcome is contrary to that of Virginia et al. (2014), who found o-phosphocholine increased from early to mid-lactation; however, unlike in our study, cows were fed a TMR-based dietary regime [35]. Collectively, our results further highlight the metabolomic signature associated with early-lactation milk when compared to both mid- and late-lactation milk. Moreover, this metabolomic approach provided a comprehensive overview of alterations occurring in the amino and organic acid levels of the bovine milk metabolome across lactation.

While the present work has identified key metabolite changes across lactation in samples from a pasture-based system, further work is needed to examine changes in lactation in different dietary regimes. The work should be expanded to examine downstream products and their metabolite composition and variation across lactation. Future work should also link the changes in metabolite composition across lactation with metabolic parameters from the cows. Establishing if dietary interventions could shift the metabolic profile will be important going forward. Importantly, using both NMR and LC-MS approaches allowed comprehensive coverage of the milk metabolome. Although there were some overlapping metabolites, both approaches yielded unique information.

## 4. Materials and Methods

### 4.1. Milk Powder Samples

Briefly, bovine milk powder samples (N = 18) from early, mid and late lactation were obtained from Dairygold Co-Operative Society Limited, Clonmel Road, Mitchelstown, Co. Cork, P67 DD36, Ireland. Dairgold is a farmer-owned co-operative based in the south of Ireland that collects milk locally in the region. Milk powder samples were spread evenly across lactation and were obtained from cows fed a natural, pasture-based diet. All samples were medium-heat-processed milk powders, manufactured through spraying drying and pasteurization, and therefore remained similar in terms of nutritional quality to fresh milk samples. 

### 4.2. Experimental Design

Metabolomic analysis of the milk powder was performed using nuclear magnetic resonance (^1^H-NMR) spectroscopy and liquid chromatography tandem–mass spectrometry platforms (LC-MS/MS). Using a multiplatform approach combining NMR and LC-MS/MS resulted in a broad coverage of various metabolites and compound classes. The dataset was composed of samples across different lactation stages: early, mid and late (N = 18). Reconstitution of the milk powder was achieved by adding 10 mL of 37 °C high-performance liquid chromatography (HPLC)-grade H_2_O to 1 g of sample, with resulting milk samples shaken at room temperature for an hour.

### 4.3. Gross Compositional Analysis

A subset of milk powder samples across early, mid and late lactation were selected at random from gross compositional analysis (N = 6). Milk macro- and micronutrients such as milk fat, minerals, vitamins and amino acids were analysed. Analysis of amino acids, vitamins and minerals was completed by ALS Life Science, Carrigeen Business Park, Clonmel, Co. Tipperary.

### 4.4. Metabolomic Analysis

#### 4.4.1. ^1^H-NMR Sample Preparation and Analysis

Following reconstitution of the bovine milk powder (1 g of sample in 10 mL of 37 °C HPLC-grade H_2_O, reconstituted milk samples (N = 18) were centrifuged at 2897× *g* for 15 min at 4 °C, and the aqueous phase was removed. The 3 kDa ultra centrifugal filters (Sigma-Aldrich, Merck KGaA, Darmstadt, Germany) were washed with 500 μL of HPLC-grade H_2_O and centrifuged at 9520× *g* for 10 min. This procedure was repeated a total of five times. Centrifuged milk samples were filtered through washed 3 kDa ultra centrifugal filters. The sample filtrate was collected, and the filtrates were then frozen at −20 °C until further analysis. 

On the day of analysis, samples were defrosted and mixed with 10 μL of sodium trimethylsilyl [2,2,3,3-2H4] proprionate (TSP) and 60μL of deuterium oxide (D_2_O). Spectra were acquired with a 600 MHz Varian Spectrometer (Varian Limited, Oxford, UK), using the first increment of a nuclear Overhauser enhancement spectroscopy pulse sequence at 25 °C. Spectra were acquired using 16,384 complex data points and 256 scans. Water suppression was achieved during the relaxation delay (2 s) and the mixing time. All ^1^H-NMR milk sample spectra were referenced to TSP at 0.0 parts per million (ppm) and processed manually with the Chenomx NMR Suite (version 7.7) using a line broadening of 0.2 Hz, followed by phase and baseline correction. Data were normalized to the total area of the spectra integral. ^1^H-NMR spectra were exported at high resolution using 6600 spectral regions. The water region (4.0–5.5 ppm) was excluded from analysis. In total, 41 metabolites were identified based on the Chenomx Library 600 MHz Library and The Human Metabolome Database (HMBD).

#### 4.4.2. LC-MS/MS Sample Preparation and Analysis

The reconstituted milk samples (1 g of sample in 10 mL of 37 °C HPLC-grade H_2_O) were prepared for a targeted metabolomic analysis according to the MxP^®^ Quant 500 assay manual (Biocrates Life Sciences, Innsbruck, Austria). The samples were ran in a randomized order. A total of 10 μL of each sample was added to the filter inserts of the 96-well plate, and then dried under a continuous nitrogen stream for 30 min at room temperature. Following drying, 50 μL of derivatization solution was added, and the plate was then incubated for 25 min and dried under a nitrogen stream for 60 min. Metabolites were then extracted and centrifuged for 500× *g* for 2 min. HPLC-grade H_2_O (150 μL) was added to the eluate (150 μL) to perform LC-MS/MS. For flow injection analysis–tandem mass spectrometry (FIA-MS/MS), 50 μL of eluate was added to the running solvent. The resulting 96-well plate was analysed using a Sciex ExionLC series UHPLC system coupled with a Sciex QTRAP 6500+ mass spectrometer. The UHPLC column was employed, and 100% water and 95% acetonitrile (both added 0.2% formic acid) were prepared as mobile phase A and B, respectively. Amino acids (n = 20), amino acid-related compounds (n = 30), bile acids (n = 14), biogenic amines (n = 9), carboxylic acids (7), hormones and related compounds (n = 4), indoles and derivatives (n = 4), nucleobases and related compounds (n = 2), fatty acids (n = 12) and other metabolites, such as trigonelline, trimethylamine N-oxide, p-Cresol sulphate and choline were quantified using LC-MS/MS. Lipid classes such as lysophosphatidylcholines (n = 14), phosphatidylcholines (n = 76), sphingomyelins (n = 15), ceramides (n = 28), dihydroceramides (n = 8), hexosylceramides (n = 19), dihexosylceramides (n = 9), trihexosylceramides (n = 6), cholesteryl esters (n = 22), diglycerides (n = 44) and triglycerides (n = 242) were semi-quantified via FIA-MS/MS. Furthermore, acylcarnitines (n = 40) and the sum of hexose were also quantified via FIA-MS/MS analysis. The multiple reaction monitoring (MRM) method was applied to identify and quantify all of metabolites.

##### Data Processing and Metabolite Quantification

MetIDQ software provided by Biocrates Life Sciences was applied to process data. Amino acids, amino acid-related metabolites and biogenic amines were quantified using isotopically labelled internal standards and seven-point calibration curves. All other metabolites were semi-quantified using internal standards. Data quality was assessed by investigating the accuracy and reproducibility of the quality control sample with a Quant 500 assay. Metabolite levels in micromoles were exported, and any metabolites above the limit of detection (LOD) in > 75% of samples were included for statistical analyses.

### 4.5. Statistical Analysis

Univariate statistical analyses of ^1^H-NMR and LC-MS/MS data were performed using Metaboanalyst 5.0 software (www.metaboanalyst.ca (accessed on 18 August 2023)). One-way repeated measures of analysis variance (ANOVA) with the post hoc Tukey test was performed on the datasets to compare average metabolite levels across the lactation stages. *p*-values were corrected for multiple comparisons using the Benjamini and Hochberg false discovery rate (FDR) procedure (FDR < 0.05). 

Significant metabolites were interpreted on a metabolite class basis, as predefined by the MxP^®^ Quant 500 list of metabolites (www.biocrates.com (accessed on 18 August 2023)). Triglycerides were analysed based on total carbon number, with the carbon number representing the total number of acyl carbons. Similarly, triglycerides were also analysed according to total number of double bonds within the triglyceride structure. Graphs were generated using Metaboanalyst (heatmap) and Microsoft Excel (stacked bar graph and bubble graphs).

## 5. Conclusions

In conclusion, the present work identifies a significant impact of lactation on the metabolomic profile of bovine milk in a pasture-based system. These findings contribute to our knowledge of lactation stage and provide insight into its impact on specific metabolite classes. In particular, differences in triglycerides and phospholipids were observed with a decrease in triglycerides of a higher total carbon number and an increase in triglycerides with lower carbon numbers across lactation. 

Understanding the impact of external factors, such as dietary regime across a full lactation season, on the metabolomic profile of milk could further enhance insights, which would be of interest to the dairy industry. Therefore, further research is warranted to investigate the impact of alterations in metabolite levels on downstream dairy products and if animal feeding practices alter these changes across lactation. 

## Figures and Tables

**Figure 1 molecules-28-06608-f001:**
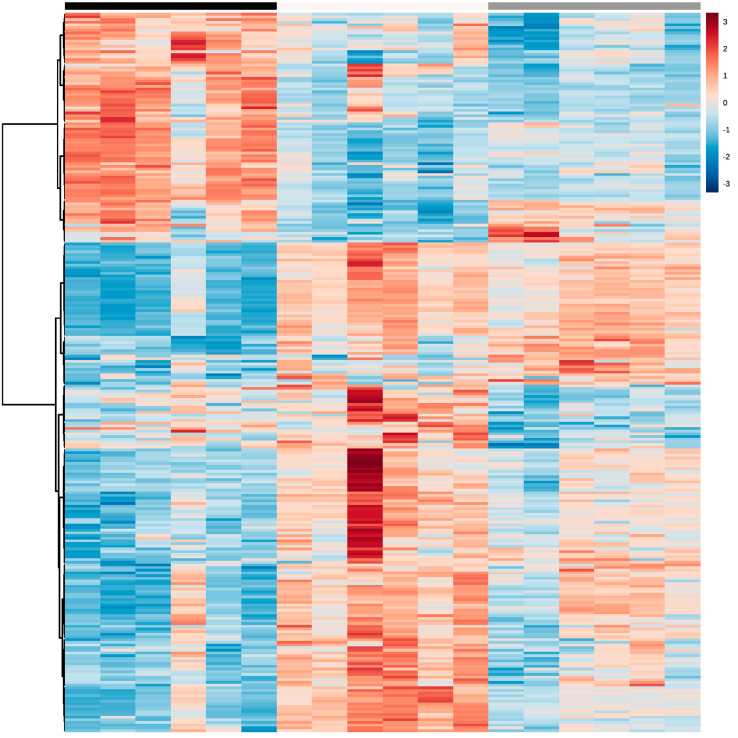
Heatmap of significant metabolites from LC-MS/MS bovine milk data for all three lactation stages: early (black), mid (white) and late (grey) (N = 18). The degree of positive or negative correlation between metabolites and lactation stage is indicated by +3 (red) to −3 (blue).

**Figure 2 molecules-28-06608-f002:**
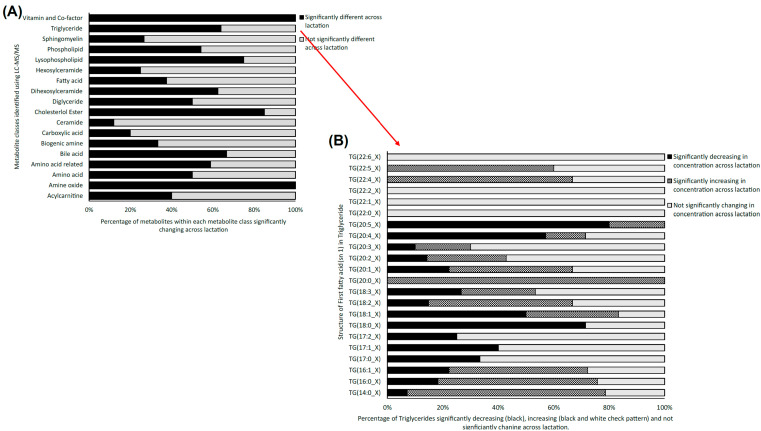
(**A**) Stacked bar chart depicting the percentage of significantly changing metabolites from the 18 metabolite classes targeted with LC-MS/MS analysis of bovine milk data across three lactation stages: early, mid and late (FDR ≤ 0.05) (N = 18). (**B**) Stacked bar chart depicting the statistically significantly increasing (grey), decreasing (black) and non-significantly changing levels of triglyceride classes from early to late lactation (FDR ≤ 0.05), as determined using LC-MS/MS analysis.

**Figure 3 molecules-28-06608-f003:**
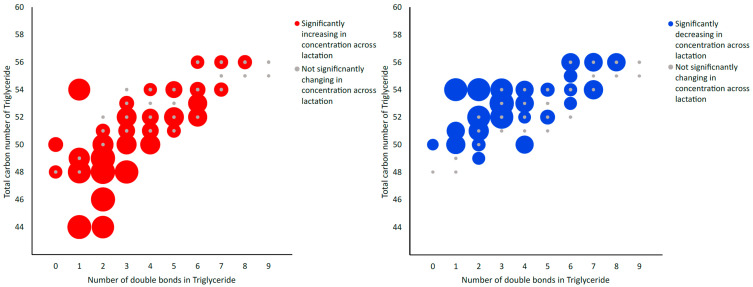
Bubble graph depicting alterations in the total triglyceride composition of milk powder samples across lactation with respect to total carbon number and number of double bonds. Larger bubbles are indicative of statistical significance (FDR ≤ 0.05), with blue bubbles indicative of decreasing concentration across lactation, red bubbles indicative of increasing concentration and grey bubbles indicative of non-significantly changing concentrations across lactation stage, as determined using LC-MS/MS analysis.

**Table 1 molecules-28-06608-t001:** Composition of milk samples in each lactation stage (N = 6).

Milk Component		Early Lactation	Mid Lactation	Late Lactation
Fat	Fat content (g/100 g)	26.2 (0.6)	25.9 (0.1)	25.3 (0.1)
	SAFA (g/100 g)	16.0 (0.4)	16.3 (0.6)	16.0 (0.1)
	MUFA (g/100 g)	7.5 (0.3)	6.4 (0.4)	6.5 (0.1)
	PUFA (g/100 g)	0.7 (0.0)	0.7 (0.2)	0.5 (0.0)
	TFA (g/100 g)	0.9 (0.0)	1.3 (0.0)	1.3 (0.0)
	Sum of Omega-3 FA (g/100 g)	0.2 (0.0)	0.2 (0.0)	0.2 (0.0)
	Sum of Omega-6 FFA (g/100 g)	0.5 (0.0)	0.5 (0.2)	0.3 (0.0)
Minerals	Ca (mg/kg)	9715.0 (35.4)	8755.0 (134.4)	8395.0 (275.8)
	CU (mg/kg)	0.7 (0.0)	0.4 (0.0)	0.2 (0.0)
	Fe (mg/kg)	1.7 (0.2)	1.4 (0.1)	1.6 (0.0)
	Mg (mg/kg)	969.5 (24.8)	862.0 (60.8)	898.5 (12.0)
	Mn (mg/kg)	0.2 (0.0)	0.2 (0.0)	0.2 (0.0)
	MO (mg/kg)	0.3 (0.0)	0.3 (0.0)	0.3 (0.0)
	P (mg/kg)	8935.0 (120.2)	8305.0 (417.2)	7465.0 (176.8)
	K (mg/kg)	11,750.0 (70.7)	11,200.0 (707.1)	10,000.0 (424.3)
	Na (mg/kg)	3085.0 (35.4)	2670.0 (127.3)	2830.0 (14.1)
	Zn (mg/kg)	33.7 (0.1)	28.6 (0.0)	27.3 (0.4)
Vitamins	Vitamin B12 (µg/100 g)	2.0 (0.0)	1.6 (0.3)	1.7 (0.1)
	Vitamin C (mg/kg)	62.6 (1.7)	83.1 (7.1)	84.5 (2.1)
Amino acids	Aspartic Acid (g/100 g)	2.4 (0.7)	1.9 (0.1)	1.9 (0.0)
	Serine (g/100 g)	1.4 (0.0)	1.4 (0.1)	1.4 (0.0)
	Glutamic Acid (g/100 g)	4.9 (0.2)	5.1 (0.2)	5.1 (0.0)
	Glycine (g/100 g)	0.8 (0.0)	0.8 (0.0)	0.9 (0.0)
	Histidine (g/100 g)	0.6 (0.0)	0.6 (0.0)	0.6 (0.0)
	Arginine (g/100 g)	0.8 (0.0)	0.8 (0.0)	0.9 (0.0)
	Threonine (g/100 g)	1.0 (0.0)	1.1 (0.1)	1.0 (0.0)
	Alanine (g/100 g)	0.8 (0.0)	0.8 (0.0)	0.8 (0.0)
	Proline (g/100 g)	2.1 (0.1)	2.2 (0.1)	2.3 (0.0)
	Cystine (g/100 g)	0.2 (0.0)	0.2 (0.0)	0.2 (0.0)
	Tyrosine (g/100 g)	1.1 (0.0)	1.1 (0.0)	1.1 (0.0)
	Valine (g/100 g)	1.1 (0.1)	1.3 (0.1)	1.3 (0.1)
	Methionine (g/100 g)	0.5 (0.0)	0.5 (0.0)	0.5 (0.0)
	Lysine (g/100 g)	2.0 (0.1)	2.1 (0.1)	2.1 (0.0)
	Isoleucine (g/100 g)	0.9 (0.1)	1.0 (0.1)	1.1 (0.1)
	Leucine (g/100 g)	2.1 (0.1)	2.3 (0.1)	2.3 (0.1)
	Phenylalanine (g/100 g)	1.1 (0.0)	1.2 (0.1)	1.2 (0.0)

Data are presented as mean (SD) of the relative concentration for milk components for each lactation stage: early (N = 2), mid (N = 2) and late (N = 2). N, number of samples; SD, standard deviation; SAFA, saturated fatty acid; MUFA, mono-unsaturated fatty acid; PUFA, polyunsaturated fatty acid; TFA, total fatty acid; CA, calcium; CU, copper; Fe, iron; Mg, magnesium; Mn, manganese; MO, molybdenum; P, phosphorus; K, potassium; Na, sodium; Zn, zinc.

**Table 2 molecules-28-06608-t002:** Average levels (mM) of metabolites in milk samples in each lactation stage (N = 18), determined via ^1^H-NMR.

Compound	Early Lactation	Mid Lactation	Late Lactation	*p*-Value	FDR
Creatinine-phosphate	0.17 (0.03) ^a^	0.11 (0.01) ^b^	0.1 (0.003) ^b^	<0.001	0.001
O-Phosphocholine	0.38 (0.09) ^a^	0.07 (0.02) ^b^	0.03 (0.01) ^b^	<0.001	<0.001
N-Acetylglucosamine	0.27 (0.07)	0.33 (0.1)	0.24 (0.09)	0.369	0.532
Creatine	0.66 (0.06) ^a^	0.51 (0.6) ^b^	0.50 (0.03) ^b^	<0.001	0.003
Citrate	6.44 (0.49) ^a^	5.11 (0.75) ^b^	4.71 (0.45) ^b^	<0.001	0.001
Choline	0.21 (0.02) ^a^	0.31 (0.04) ^b^	0.34 (0.02) ^c^	<0.001	<0.001
Alanine	0.03 (0.003)	0.03 (0.01)	0.03 (0.004)	0.44	0.565
Dimethylamine	0.14 (0.02) ^a^	0.12 (0.02) ^b^	0.12 (0.02) ^b^	0.003	0.012
Glutamate	0.27 (0.03) ^a^	0.25 (0.04)	0.22 (0.02) ^b^	0.007	0.023
3-Hydroxybutyrate	0.04 (0.01) ^a^	0.03 (0.01) ^b^	0.03 (0.01) ^b^	0.007	0.023
Acetone	0.09 (0.02)	0.09 (0.03)	0.1 (0.04)	0.848	0.848
Glucose-1-phosphate	0.14 (0.03) ^a^	0.03 (0.01) ^b^	0.04 (0.05) ^b^	<0.001	<0.001
Butyrate	0.04 (0.01)	0.04 (0.01)	0.05 (0.01)	0.388	0.533
2-Oxoglutarate	0.12 (0.02)	0.12 (0.02)	0.12 (0.01)	0.764	0.802
Isovalerate	0.02 (0.01)	0.01 (0.01)	0.03 (0.01)	0.139	0.278
Glucose-6-phosphate	0.47 (0.18)	0.3 (0.09)	0.37 (0.18)	0.154	0.295
Glycerophosphocholine	7.00 (1.18) ^a^	6.63 (0.66) ^b^	6.12 (0.40) ^b^	0.002	0.012
Hippurate	0.15 (0.03) ^b^	0.19 (0.02) ^a^	0.17 (0.02) ^b^	0.009	0.025
Isoleucine	0.00 (0.001)	0.00 (0.002)	0.01 (0.001)	0.305	0.463
Dimethyl Sulfone	0.03 (0.004) ^b^	0.04 (0.003) ^a^	0.03 (0.003) ^b^	<0.001	0.002
Valerate	0.16 (0.01)	0. 02 (0.004)	0.03 (0.01)	0.111	0.232
Fumarate	0.02 (0.002)	0.02 (0.003)	0.02 (0.002)	0.172	0.315
Glycerol	8.16 (1.05)	8.63 (0.66)	8.33 (1.4)	0.834	0.848
Glucose	0.25 (0.07)	0.32 (0.12)	0.21 (0.11)	0.195	0.342
Valine	0.01 (0.001)	0.01 (0.002)	0.01 (0.002)	0.595	0.694
Pyruvate	0.02 (0.01)	0.02 (0.003)	0.02 (0.004)	0.393	0.533
Succinate	0.04 (0.003)	0.04 (0.01)	0.04 (0.01)	0.733	0.791
Galactose	0.21 (0.13) ^a^	0.39 (0.12) ^b^	0.36 (0.10) ^b^	0.010	0.027
Lactate	0.23 (0.21)	0.63 (0.84)	0.81 (0.71)	0.309	0.463
Formate	0.09 (0.07)	0.26 (0.35)	0.34 (0.33)	0.305	0.463
Lactose	67.02 (11.36)	70.14 (9.40)	69.78 (5.96)	0.540	0.648
Taurine	0.09 (0.01)	0.09 (0.03)	0.08 (0.03)	0.468	0.578
Ethanolamine	0.05 (0.01) ^a^	0.08 (0.04) ^b^	0.09 (0.02) ^b^	0.016	0.039
Glycine	4.72 (0.48)	4.88 (0.44)	4.59 (0.78)	0.733	0.791

Data are presented as mean (SD) of the relative levels for compounds for each lactation stage: early, mid and late. N, number of samples; SD, standard deviation; false discovery rate, FDR; determined using ANOVA. FDR < 0.05 was considered significant. a, b, c: different superscript letters represent significant differences between lactation stages (FDR < 0.05).

## Data Availability

The data presented in this study are available on request from the corresponding author.

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
