# Peer review of "Impact of Lactation Stage on the Metabolite Composition of Bovine Milk"

_molecules, 2023, doi:10.3390/molecules28186608_

Round 1

Reviewer 1 Report

This research is interesting and the results can provide data support for nutritional composition of bovine milk across the whole lactation stage. Meanwhile, the finding of this work can provide theoretical support for the bovine milk processing. However, there are some questions should be illustrated.

1. There are some format errors in the manuscript (such as line 244, line 260) need to be modified.

2. The state of bovine milk is powder or liquid? Please clarify.

3. In the part of “Materials and Methods”, the author claimed the sample used in this research is milk powder. Dose the process stage will change the substance composition of milk powder when compared with fresh milk.

4. Please confirm whether the value of “0.0 parts per million (ppm)“ is correct in line 257.

5. The conclusion do not obtain details of the results. The author should rewrite it more detailed.

Due to the paper have some format and grammar errors, the author need take minor editing of this paper.

Reviewer 2 Report

In this article, nuclear magnetic resonance (1H-NMR) spectroscopy and liquid chromatography tandem mass spectrometry (LC-MS/MS) were used to examine the impact of lactation stage on bovine milk metabolite levels. Compound classes significantly affected by lactation stage included phosphatidylcholines (59%) and triglycerides (64%), of which 100% phosphatidylcholine and 61% of triglycerides increased from early lactation onwards.

1.        In the introduction section, the author declared that “Although several studies examine alterations in the metabolomic profile of milk and dairy products concerning parity, diet, and lactation, the true impact of lactation in a pasture based dietary system is not fully understood.” Please give the conclusion that, in this research this question was fully understood or not.

2.        A multiplatform metabolomic approach (NMR and LCMS) was used in this research. Please explain why these two approaches were chosen from a technical and outcome perspective.

3.        532 metabolites were detected in NMR method and 491 metabolites were detected in LCMS method. Please analyze the similarities and differences of these metabolites, and analyze the need for the combination of the two methods based on these results.

4.        Compounds quantification was carried out in this article. Was this method validated before quantification? Please provide methods and data of method validation.

Minor editing of English language required.
